# Tribological Properties of Molybdenum Disulfide and Helical Carbon Nanotube Modified Epoxy Resin

**DOI:** 10.3390/ma12060903

**Published:** 2019-03-18

**Authors:** Zhiying Ren, Yu Yang, Youxi Lin, Zhiguang Guo

**Affiliations:** 1College of Mechanical Engineering and Automation, Fuzhou University, Fuzhou 350116, China; renzyrose@126.com (Z.R.); yangflyyu@163.com (Y.Y.); 2State Key Laboratory of Solid Lubrication, Lanzhou Institute of Chemical Physics, Chinese Academy of Sciences, Lanzhou 730000, China

**Keywords:** MoS_2_, helical carbon nanotubes, epoxy resin, tribological properties

## Abstract

In this study, epoxy resin (EP) composites were prepared by using molybdenum disulfide (MoS_2_) and helical carbon nanotubes (H-CNTs) as the antifriction and reinforcing phases, respectively. The effects of MoS_2_ and H-CNTs on the friction coefficient, wear amount, hardness, and elastic modulus of the composites were investigated. The tribological properties of the composites were tested using the UMT-3MT friction testing machine, non-contact three-dimensional surface profilometers, and nanoindenters. The analytical results showed that the friction coefficient of the composites initially decreased and then increased with the increase in the MoS_2_ content. The friction coefficient was the smallest when the MoS_2_ content in the EP was 6%, and the wear amount increased gradually. With the increasing content of H-CNTs, the friction coefficient of the composite material did not change significantly, although the wear amount decreased gradually. When the MoS_2_ and H-CNTs contents were 6% and 4%, respectively, the composite exhibited the minimum friction coefficient and a small amount of wear. Moreover, the addition of H-CNTs significantly enhanced the hardness and elastic modulus of the composites, which could be applied as materials in high-temperature and high-pressure environments where lubricants and greases do not work.

## 1. Introduction

Epoxy resin (EP) has excellent properties such as strong adhesion, high mechanical properties, and good dimensional and chemical stabilities. It is a thermosetting resin and widely used in coatings, adhesives, and composite materials [1,2]. However, the cured three-dimensional cross-linked structure makes EP highly brittle with a high wear rate [3]. In order to improve these performance defects and impart better properties comprehensively, it is necessary to modify EPs either physically or chemically [4,5]. Using fillers is a commonly used method for the physical modification of the EPs [6,7]. The mechanical properties of EP composites can be improved by adding fillers that improve their tribological properties. The commonly used inorganic fillers include inorganic solid lubricating materials such as molybdenum disulfide (MoS_2_), carbon fibers, and carbon nanotubes (CNTs) [8,9,10,11,12]. Molybdenum disulfide is a layered inorganic solid lubricating material. The force between the layers in its structure is small and layer sliding occurs relatively easily [13,14]. The friction coefficient and wear rate of the composites can be reduced by filling MoS_2_ in the EP. Carbon fiber and CNT fillers have high strength and stiffness, and are often used to strengthen EP to obtain excellent mechanical properties [15,16]. With the ever-increasing applications of EPs, studies relating to the improvement of their friction and wear properties by using carbon fiber fillers have attracted considerable attention. Srivastava et al. [17] modified EP with MoS_2_ and found that the coefficient of friction and wear resistance show improvements of 60% and 78%, respectively compared to the neat resin matrix. Ranjbar et al. [18] reinforced EP with carbon fibers and the results show that compared to pure epoxy Young’s modulus and tensile strength of epoxy/MWCNTs at 0.34 wt.% of CNTs are increased for 21.98% and 58.32%, respectively, and for epoxy/fiberglass/MWCNTs at 0.17 wt.% of CNTs raised for 1.05 and 1.17 times, respectively. Luangtriratana et al. [19] filled EP with CNTs and studied the effect of CNT content and type on the friction and wear properties of the EP composites. 

Based on available literature, it is evident that not only do helical carbon nanotubes (H-CNTs) have excellent properties of CNT, but also would have even better properties as that filled with CNTs [20,21], because of their unique spiral structure. Therefore, a modified EP composite filled with H-CNTs would have a better effect. To date, H-CNTs and MoS_2_ have not been used together as additives to improve the friction properties of EP. In this study, EP, MoS_2_, and H-CNTs were employed as the matrix, anti-wear phase, and reinforcing phase, respectively, to prepare several composite materials. The friction and wear properties of the composite materials were investigated. It was anticipated that these composite materials would exhibit low friction coefficient, small wear, and high-elastic modulus.

## 2. Materials and Methods 

### 2.1. Materials

Table 1 and Figure 1 outline the various materials used for the experiments and their specific parameters.

### 2.2. Sample Preparation

#### 2.2.1. Pre-Treatment of H-CNTs

H-CNTs have a non-polar surface, which barely interacts with the EP matrix. The pure H-CNTs in the as-delivered state exemplify the tendency of MWCNTs to form a kind of loosely packed felt. The pores inside these non-polar agglomerates are so small (in the range of a few 100 nm) that the polar EP resin cannot easily infiltrate them [22]. Some H-CNT samples were therefore pre-treated to improve the bonding between H-CNTs and matrix. The certain amount of H-CNTs was placed in a mixture of concentrated H_2_SO_4_ and concentrated HNO_3_ with a volume ratio of 3:1, then dispersed in an ultrasonic cleaner for 30 min at room temperature. Next, the ultrasonically dispersed H-CNTs was transferred to a one-necked flask and refluxed for 3 h at 120 °C in a constant temperature oil bath. After the H-CNTs were cooled to room temperature, they were transferred to a beaker, centrifuged at high speed, and then vacuum dried at 80 °C for 5 h to obtain acid-modified H-CNTs. Figure 2 shows the field emission scanning electron microscopy images (FESEM, JSM-6701F, JEOL, Tokyo, Japan) of H-CNTs and acid-H-CNTs. Compared with the original H-CNTs, many polar functional groups, including –COOH, –OH and –CO–, were introduced into the wall of the acid-H-CNTs. Therefore, the acid-H-CNTs can be well-combined with the EP, and the dispersion of H-CNTs in epoxy resin is also improved.

#### 2.2.2. Synthetic Process

The synthetic process of the EP composites was as follows. A certain amount of the EP was added to a beaker, which was heated to 60 °C to reduce the viscosity of the epoxy resin. Next, specific proportions of MoS_2_ and acid-modified H-CNTs were added to the reaction vessel and the contents were stirred using a magnetic stirrer for 30 min, followed by shaking for another 30 min to disperse MoS_2_ and acid-modified H-CNTs evenly in the epoxy resin. Subsequently, the curing agent was added proportionally, and after stirring for 30 min, the reaction vessel was put into the vacuum drying box for another 30 min to remove the matrix bubbles and water. The mixture was then poured into a silica gel mold that was coated with a demoulding agent and cured at room temperature for 24 h. Later, the sample was removed from the mold and sanded to a level with 400, 1200, 5000 mesh sandpaper.

### 2.3. Microstructure of the Composites 

Figure 3 shows the micro-morphology of (a) pure EP, (b) EP + 6%MoS_2_, (c) EP + 4%H-CNTs, and (d) EP + 6%MoS_2_ + 4%H-CNTs, respectively. It can be seen from Figure 3 that the additives were bound well with the EP, and the curing and crosslinking of the EP were uniform. In order to observe the distribution uniformity of the additives in the EP, the distribution of elements in the 6% MoS_2_ and 4% H-CNTs composites was studied by energy dispersive spectroscopy (EDS, JSM-5600LV, JEOL, Tokyo, Japan) (Figure 4). As each element was evenly distributed in the spectrum, it was clear that the additives were evenly dispersed in the EP. Since the two additives are evenly distributed in the epoxy resin, each additive will be evenly distributed when added separately.

### 2.4. Test Method

The friction coefficient of the composites was tested by the UMT-3MT friction tester (Bruker Company, Hamburg, Germany). According to the mechanical properties of the EP, the width and depth of wear traces were assessed, and the optimum scratch parameters were determined by debugging several times. The GCr15 ball with a diameter of 5 mm was used as counter-body, the loads on the x- and z-axes were 3 N and 5 N, respectively, the loading time was 5 s, and the motion mode was high-speed linear reciprocating motion. The velocity, distance of sliding, and test time were 5 mm/s, 8 mm, and 300 s, respectively. The test time was initially set to 30 min, but it was observed that the coefficient of friction will reach a plateau after around 50 s, and afterwards it will change very little, therefore the test time was set at 300 s.

Specific test methods: First, the instrument parameters were set. The sample was then clamped to ensure that its surface remained on the level. Next, the ball was adjusted to be in the vicinity of the sample surface, and finally, the instrument was run. When the counter-body detected the sample surface, the load gradually increased to the peak load, loading time was 5 s, and then the counter-body started to move back and forth at the set speed until the end of the test. At this time, the test data were saved and the friction coefficient curve was fitted. In order to ensure the accuracy of the data, each sample was measured three times at different positions, and then the average value after the friction entered a stable state was calculated to obtain the average friction coefficient. Figure 5 shows a physical view of the three main test instruments.

## 3. Results

### 3.1. Friction Coefficient

In order to compare the effects of additives MoS_2_ and H-CNTs, and their interaction on the friction properties of epoxy resin composites, three groups of experiments were conducted. In these experiments, 2%, 4%, 6%, and 8% MoS_2_, and H-CNTs were added to the EP. Then, the optimum MoS2 concentration was selected, and 2%, 4%, 6%, and 8% H-CNTs were added to the composites. Since the friction coefficient of the composite material is the lowest when the MoS_2_ content is 6%, the optimum content of MoS_2_ is 6%. The friction coefficients of the composites with different contents of various additives are shown in Table 2 and Figure 6. The friction coefficient in Table 2 is the average value of all the data after the test enters a stable state, which is also the average value of all the data after the 50 s.

The data in Table 2 and Figure 6a, reveals that the friction coefficient of the EP composites decreased gradually with the increasing MoS_2_ content. As MoS_2_ is a typical layered structure and the interaction force between the layers is very weak (Van der Waals force), the layers can slip easily, thus reducing the friction coefficient of the friction surface. When the content reached 6%, the friction coefficient was the smallest, and the most stable. This is because with the increase of MoS_2_ content, the filler was distributed more on the friction surface. When the content of MoS_2_ exceeds 6%, the friction surface of MoS_2_ is saturated, and the friction coefficient increases. From Table 2 and Figure 6b, it is evident that the friction coefficient of composites with the H-CNTs is lower than that of pure epoxy resin, but with the increase of the H-CNTs content, the friction coefficient of composites increases gradually. As we all know, H-CNTs have distinctive seamless drums assembled by graphite sheets, which have excellent tribological properties of graphite sheets [23,24,25]. The graphite sheet is an excellent solid lubricant, so the H-CNTs exposed to the sliding interface were acting as a solid lubricant at the sliding interface, which reduced the coefficient of friction [26]. Due to the poor dispersibility of H-CNTs, as the content increases, the H-CNTs will agglomerate, the anti-friction effect will be worse, and the friction coefficient will become larger [27]. From Table 2 and Figure 6c, it can be seen that the friction coefficient of the composites with 6% MoS_2_ and gradually increasing H-CNTs decreased initially and then tended to stabilize. The main reason for this process is MoS_2_, whose antifriction principle is the same as described above. Figure 6d also revealed that for more than 2% of additives the friction coefficient of the composites with MoS_2_ was smaller than that of composites with the H-CNTs. The friction coefficient of the composites with 6% MoS_2_ and different contents of H-CNTs lay between these values, which is closer to that of the composites with MoS_2_. This showed that the friction coefficient of the composites could be reduced by all three factors, but with MoS_2_ it was the most obvious.

### 3.2. Wear Volume

In order to further illustrate the wear resistance of the composites formed by different additives and epoxy resin, the 3D wear profiles and volume of the three groups of samples were measured by a non-contact 3D surface profiler, as shown in Figure 7, Figure 8 and Figure 9. Figure 7 shows the Wear volume curve and 3D wear profiles of the original EP and EP composites with 2%, 4%, 6% and 8% MoS_2_ added, respectively. It was observed that with an increase in the MoS_2_ content, the wear loss of the composites decreased initially and then increased. The wear loss was the smallest when the content was 2%, and when the content was 8%, the wear loss was close to that of the original EP. When the content of MoS_2_ was increased, the van der Waals force between the layers was very small and they were easy to peel off, which led to an increase in the amount of wear.

Figure 8 shows the Wear volume curves and 3D wear profile of the original EP and the EP composites with 2%, 4%, 6%, and 8% H-CNTs. It was found that the wear rate of the composites initially decreased with the increasing H-CNT content and then increased at more than 4% of H-CNTs. This is because the H-CNTs have structural integrity and stronger chemical bonds, which impart excellent mechanical properties and excellent anti-deformation ability [22]. Therefore, the addition of H-CNTs into the EP matrix enhances the wear resistance of the composites. When the content of H-CNTs was more than 4%, the wear volume began to increase sharply. This could be attributed to the increased friction coefficient of the composites and thus greater adhesion when its content was too high.

Figure 9 shows Wear volume curves and 3D wear profiles of the original EP and the EP composites with 6% MoS_2_ and 2%, 4%, 6%, and 8% H-CNTs. Wear rate initially decreased with H-CNTs content and then increased at more than 6% of H-CNTs. When the content of H-CNTs was 4%, the wear rate was the smallest. But with the increase in H-CNTs content, the fillers might be prone to agglomerate, the volume of resin layer between in the composites decreases, and the binding force between epoxy and fillers weaken, leading to an increase in wear [28,29]. In conclusion, the addition of H-CNTs significantly reduced the wear of the composites, while the addition of MoS_2_ had a less pronounced effect on wear.

### 3.3. Comparison of Wear and Friction Coefficients of Composite Materials with Different Additives

Figure 10 shows the variation curves of the friction coefficient and wear rate of the composites with different additives. From Figure 10a,b, it can be concluded that MoS_2_ reduced the friction coefficient of the material without significantly improving the wear rate, and the H-CNTs reduced the wear rate of the material and did not significantly improve the friction coefficient. Therefore, the EP matrix composites with MoS_2_ and H-CNTs were prepared, so that they could impart their respective advantages and realize composites with low friction coefficient and wear rate. From the previous analysis, it was evident that the friction coefficient of the composites with different amounts of H-CNTs did not fluctuate very much, but when the content of MoS_2_ was 6%, the friction coefficient of the composites was the lowest. Therefore, the content of MoS_2_ was fixed at 6% in this experiment, and then different proportions of H-CNTs were added to test the friction coefficient and wear quantity of the composites. Figure 10c is the variation curve of the friction coefficient and wear rate of EP composites with 6% MoS_2_ and different content of H-CNTs. Figure 10c shows that with the increase in H-CNTs content, the friction coefficient of the composites fluctuated slightly and remained stable, while the wear loss first decreased and then increased, resulting in the smallest wear loss when the content of H-CNTs was 4%. Finally, the composite material with 6% MoS_2_ and 4% H-CNTs provided minimum friction and wear.

### 3.4. Wear Surface Analysis

In order to study the wear mechanism, FESEM images of the wear tracks of the composite material with pure EP and the optimal content of each additive were taken, as shown in Figure 11. Figure 11a shows the FESEM images of the worn surface of the pure EP. The remnants of a high number of holes formed by the exfoliation of the pure EP, and cracks that are perpendicular to the sliding direction are clearly visible on the worn surfaces in Figure 11a. The results show that the pure EP has obvious adhesive wear and fatigue wear. Figure 11b shows the FESEM images of the worn surface of composite (EP + 6%MoS_2_). It can be seen that cracks perpendicular to the sliding direction are still present on the wear track, but there are significantly fewer wear debris. This is because the layered structure of MoS_2_ reduces the friction coefficient and thus reduces the wear debris, but the composite still has obvious adhesive wear and fatigue wear. Figure 11c shows the FESEM images of the worn surface of composite (EP + 4%H-CNTs), where furrows parallel to the sliding direction, and the cracks perpendicular to the sliding direction disappeared. However, it was obvious that abrasive wear occurred in the composite. This is because the excellent mechanical properties of the H-CNTs enhance the tensile strength of the composite and prevent from the occurrence of fatigue fracture, while the appearance of furrows is also because the tensile strength of the composite is enhanced, therefore the grinding debris cannot be peeled off smoothly, and abrasive wear occurs in the sliding direction. Figure 11d shows the FESEM images of the worn surface of composite (EP + 6%MoS_2_ + 4%H-CNTs). Obviously, the wear marks of the composite material are smooth and the plough marks are few, indicating that the worn surface of the composite is less damaged. The synergistic effect of the H-CNTs and MoS_2_ makes the worn polymer region smoother than pure EP, and the composite exhibits slight adhesive wear and abrasive wear. The presence of H-CNTs greatly improves the strength and toughness of the composite, inhibits the generation and propagation of cracks, and prevents a large number of surface shedding of the composite; the lubricating property of MoS_2_ reduces the friction on the worn surface and greatly reduces the friction coefficient on the worn surface.

### 3.5. Modulus of Elasticity and Hardness

The above contents verify that the friction coefficient and wear rate of the EP composites with MoS_2_ and H-CNTs could be improved to a great extent. The hardness and elastic modulus of the composites have not been discussed yet. However, not only do H-CNTs have the strong tensile properties of CNTs, but also a unique spiral structure that makes them elastic. This means that the elastic modulus of the composites could be increased. In order to verify this hypothesis, the hardness and modulus of elasticity of the composites were measured by Anton Paar’s ultra-nanoindentation tester (UNHT). Among the composites, EP composites with 6% MoS_2_, and 6% MoS_2_ and 4% H-CNTs had small friction coefficients and low wear loss, therefore they were selected. The elastic modulus and hardness of these two excellent EP composites were compared with those of the original EP. 

In order to ensure the accuracy of the data, nanoindentation tests were carried out at five different locations of the same sample. Five sets of loading and unloading curves were obtained for each sample, as shown in Figure 12a. The hardness and modulus of elasticity at five different locations of each sample were obtained by using the standard linear loading mode and O&P theory [30]. Their average, standard, and median values were also calculated, as shown in Table 3. The hardness and elastic modulus were obtained by calculating the elastic modulus and hardness of materials proposed by Legocka et al. [31], Oliver and Pharr [32]. The formula for calculating the equivalent modulus $E_{r}$ and hardness *H* as:$$E_{r}=\frac{\sqrt{\pi }}{2}\frac{S}{\sqrt{A}}$$
$$H=\frac{P_{max}}{A}$$
where S is the stiffness of instantaneous unloading; A is the contact projection area product; $P_{max}$ is the peak load of the indentation process. The calculation formula of contact depth is as follows:$$h_{c}=h_{max}-\varepsilon \frac{P_{max}}{S}$$where $h_{max}$ is the maximum depth of indentation process; ε is the shape coefficient of the pressure head. For the conical pressure head, ε = 0.75. The size of $E_{r}$ is related to the elastic modulus and Poisson’s ratio of samples and head:$$E_{r}={\left(\frac{\left(1-v^{2}\right)}{E}+\frac{\left(1-v_{i}{}^{2}\right)}{E_{i}}\right)}^{-1}$$where E and v are the elastic modulus and Poisson’s ratio of the sample, respectively; $E_{i}$ and $v_{i}$ are the elastic modulus and Poisson’s ratio of diamond head, respectively. For diamond, $E_{i}$ = 1141 GPa and $v_{i}$ = 0.07.

In order to make the results more intuitive and comparative, the average value of each group of data was calculated and depicted in a histogram, as shown in Figure 12b. 

From Table 3 and Figure 13, it can be seen that the hardness and modulus of elasticity of the EP composites with 6% MoS_2_ and 4% H-CNTs are 712 MPa and 29.3 MPa, respectively, which are higher than those of the other two samples. The hardness and modulus of pure EP and EP composites with 6% MoS_2_ were close to each other, i.e., 536 MPa and 518 MPa, 17.4 MPa, and 17.6 MPa, respectively. Thus, it is evident that H-CNTs can improve the elastic modulus and hardness of the composites. In order to explore the mechanisms of improvement of elastic modulus of the EP composite with H-CNTs, the fracture surface morphology of the composites was observed by field emission scanning electron microscopy (FESEM), as shown in Figure 13. From the electron microscopic images, it is clear that the H-CNTs tightly pulled together the fractured surface in the cracks of the composites, thus increasing their elastic modulus. 

## 4. Conclusions

Epoxy resin composites were prepared with MoS_2_ and H-CNTs as the anti-wear and reinforcing phases. The effects of MoS_2_ and H-CNTs on the friction coefficient, wear rate, hardness, and elastic modulus of the composites were studied and the following conclusions were drawn:(1)When only MoS_2_ was added, the friction coefficient decreased gradually with the increasing content of MoS_2_. When the content of MoS_2_ reached 6%, the friction coefficient was the smallest, but the wear rate decreased with 2% of MoS_2_ and then increased. The wear rate was the smallest when the content of MoS_2_ was 2%, and close to that of the original epoxy resin for 8% MoS_2_ content. When only H-CNTs were added, and with the increasing content of H-CNTs, friction coefficient of the composites decreased first (for 2%), then stabilized, and subsequently increased rapidly (for more than 4%), while the wear rate decreased up to 4% of H-CNTs and then increased.(2)MoS_2_ and H-CNTs were added simultaneously to prepare the epoxy resin composites, which can reduce the friction coefficient and wear rate. By testing different contents of various additives, it became evident that the friction coefficient and wear amount of the composites reached the minimum when the contents of MoS_2_ and H-CNTs were 6% and 4%, respectively.(3)The hardness and modulus of the elasticity of three typical specimens were measured at five different locations by using the linear loading mode and O&P theory to obtain the average value. It was observed that the addition of H-CNTs enhanced the hardness and modulus of elasticity of the composites. The hardness changed from 530 MPa to 712 MPa, and the modulus of elasticity changed from 17.4 MPa to 29.3 MPa.

In summary, the composites with low friction coefficient, low wear rate, and high elastic modulus prepared in this study may be used as materials in high-temperature and high-pressure environments where lubricants and greases cannot work.

## Figures and Tables

**Figure 1 materials-12-00903-f001:**
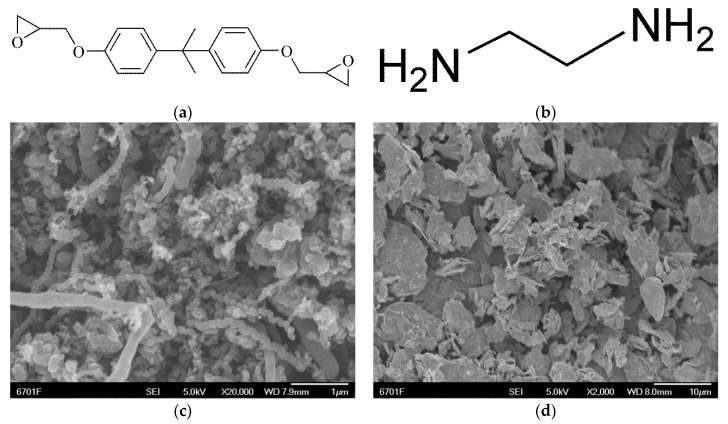
Molecular structure of (**a**) diglicidyl ether, (**b**) ethylenediamine and FESEM images of (**c**) helical carbon nanotubes (H-CNTs), (**d**) molybdenum disulfide (MoS_2_).

**Figure 2 materials-12-00903-f002:**
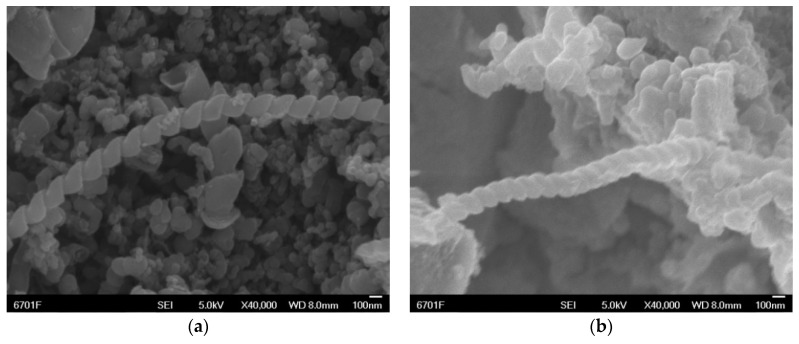
FESEM images of (**a**) H-CNTs, (**b**) acid-H-CNTs.

**Figure 3 materials-12-00903-f003:**
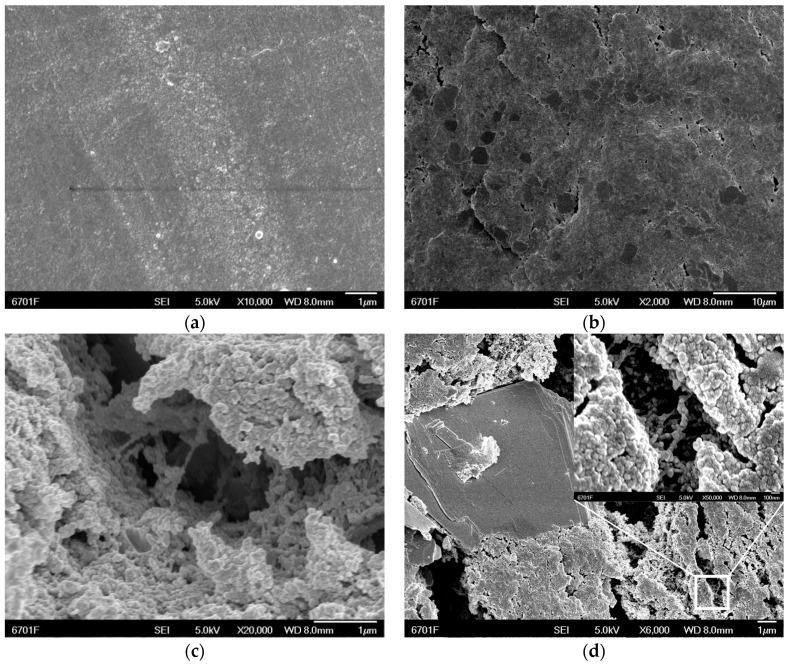
Surface topography of additives and composites. (**a**) Epoxy resin (EP), (**b**) EP + MoS_2_, (**c**) EP + H-CNTs, and (**d**) EP + MoS_2_ + H-CNTs.

**Figure 4 materials-12-00903-f004:**
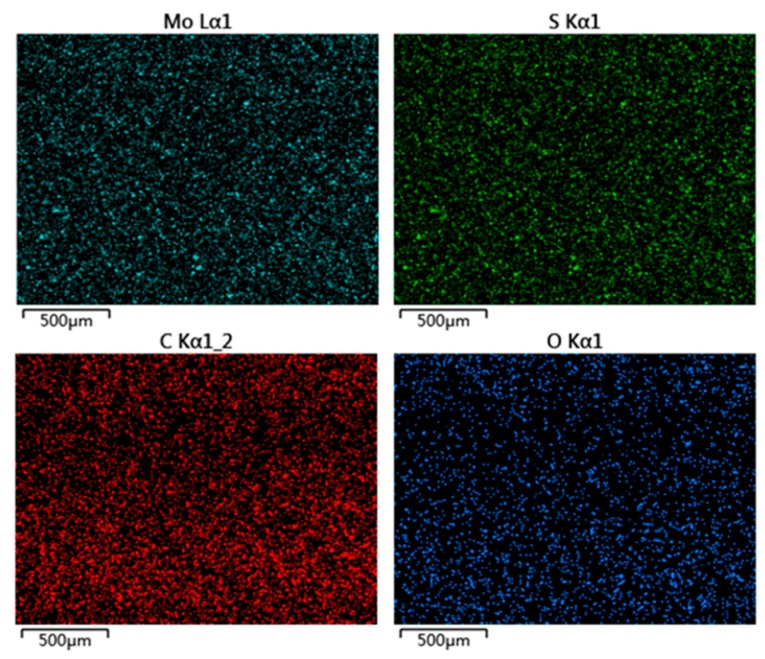
Elemental distribution maps for the surface of EP + 6%MoS_2_ + 6%H-CNTs.

**Figure 5 materials-12-00903-f005:**
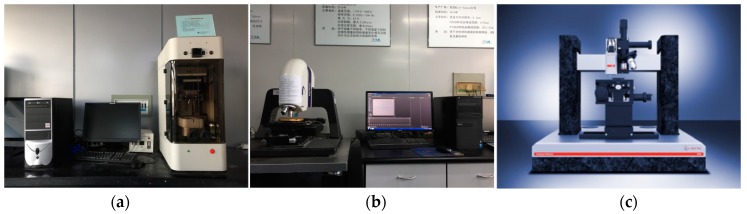
Experimental setup. (**a**) UMT-3 friction and wear testing machine, (**b**) non-contact three-dimensional surface profiler, and (**c**) ultra-nanoindenter.

**Figure 6 materials-12-00903-f006:**
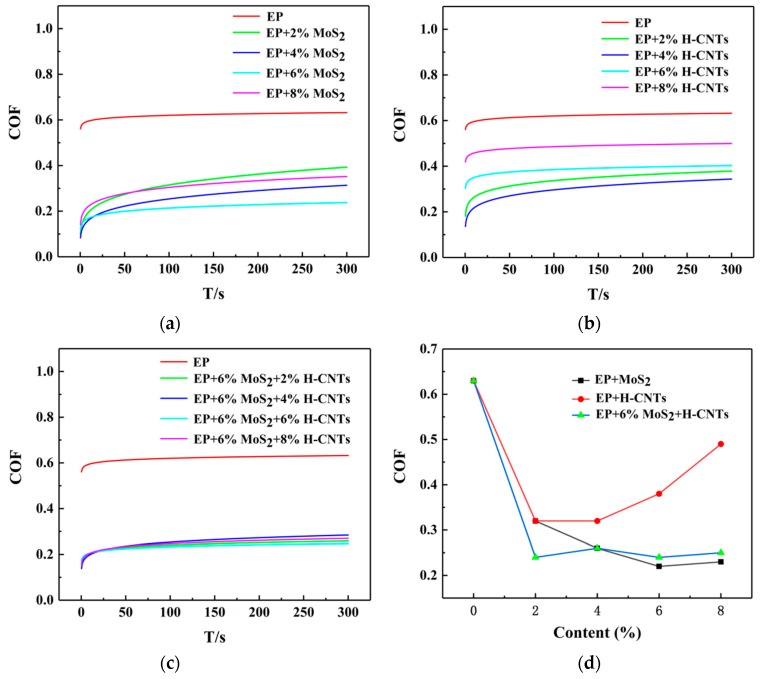
Effect of different contents of additives on the friction coefficient of the composites. (**a**) MoS_2_, (**b**) H-CNTs, (**c**) 6% MoS_2_ + H-CNTs, (**d**) comparison of the friction coefficients from three experimental groups.

**Figure 7 materials-12-00903-f007:**
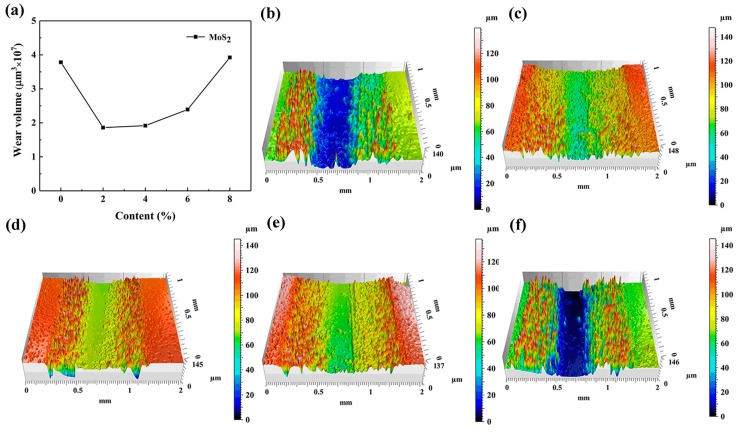
Wear volume curves and 3D wear profile of the original epoxy resin and epoxy resin composites with different MoS_2_ content. (**a**) Wear volume curve, (**b**) 0% MoS_2_, (**c**) 2% MoS_2_, (**d**) 4% MoS_2_, (**e**) 6% MoS_2_, and (**f**) 8% MoS_2_.

**Figure 8 materials-12-00903-f008:**
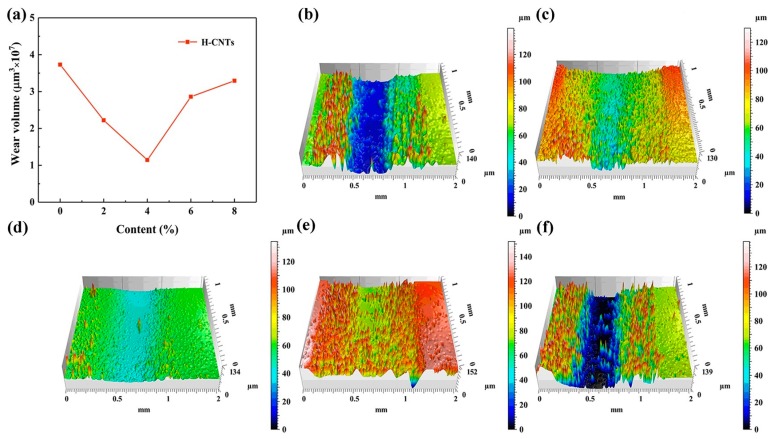
Wear volume curves and 3D wear profile of the original epoxy resin and the epoxy resin composites with different H-CNT contents. (**a**) Wear volume curve, (**b**) 0% H-CNTs, (**c**) 2% H-CNTs, (**d**) 4% H-CNTs, (**e**) 6% H-CNTs, and (**f**) 8% H-CNTs.

**Figure 9 materials-12-00903-f009:**
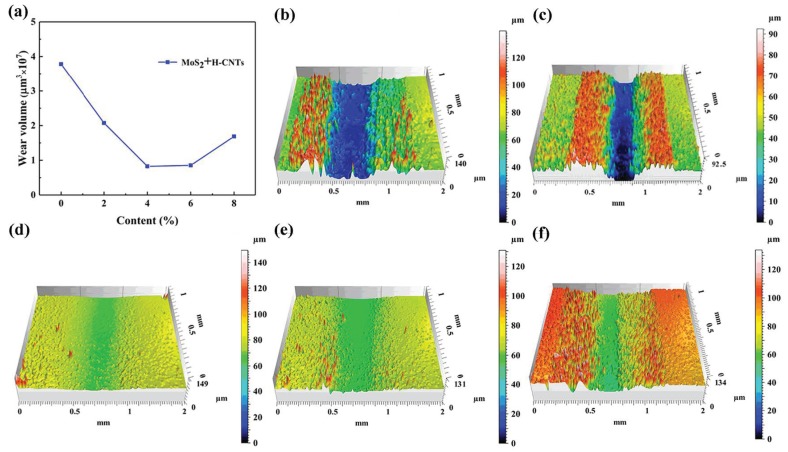
Wear volume curves and 3D wear profile of the original epoxy resin and the epoxy resin composites with different MoS_2_ and H-CNTs contents. (**a**) Wear volume curve, (**b**) 6% MoS_2_ + 2% H-CNTs, (**c**) 6% MoS_2_ + 2% H-CNTs, (**d**) 6% MoS_2_ + 4% H-CNTs, (**e**) 6% MoS_2_ + 6% H-CNTs, and (**f**) 8% H-CNTs.

**Figure 10 materials-12-00903-f010:**
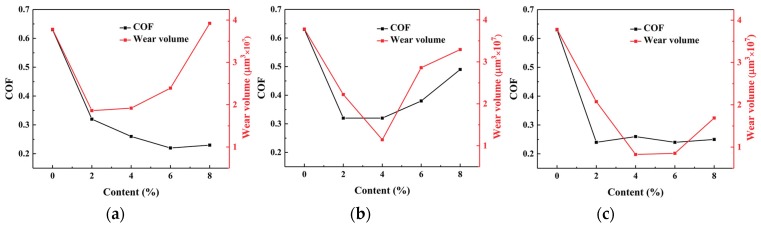
Comparison of wear and friction coefficient of composites with different additives. (**a**) MoS_2_, (**b**) H-CNTs, and (**c**) 6% MoS_2_ + H-CNTs.

**Figure 11 materials-12-00903-f011:**
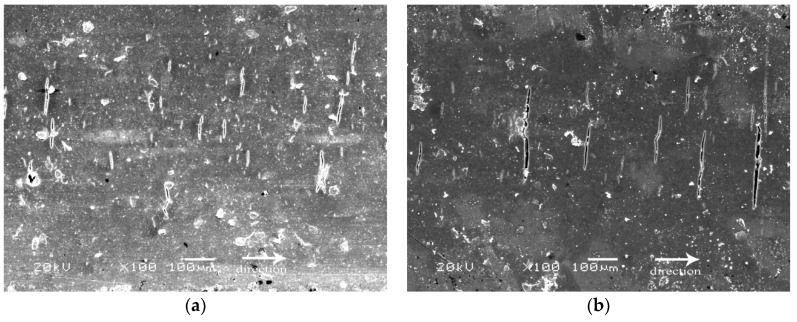
FESEM images of worn surfaces composite (**a**) EP, (**b**) EP + 6%MoS_2_, (**c**) EP + 4%H-CNTs, (**d**) EP + 6% MoS_2_ + 4%H-CNTs.

**Figure 12 materials-12-00903-f012:**
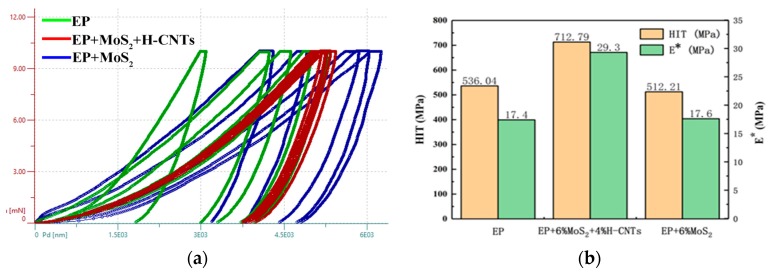
Nanoindentation test. (**a**) Loading and unloading curve. (**b**) Hardness and elastic modulus.

**Figure 13 materials-12-00903-f013:**
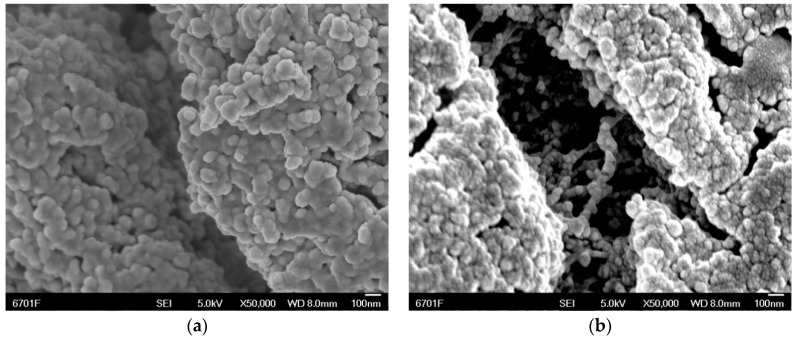
Fracture surface morphology of different composites. (**a**) EP; (**b**) EP + 6% MoS_2_ + 4% H-CNTs.

**Table 1 materials-12-00903-t001:** Experimental materials.

Component	Requirement
Epoxy resin	Diglicidyl ether
Curing agent	Ethylenediamine
H-CNTs	Diameter 100–200 nm, length 1–10 μm, carbon content more than 95%, ignition point 560–600 ℃.
MoS_2_	The content should be more than or equal to 98%.

**Table 2 materials-12-00903-t002:** Coefficient of friction of composites with different contents of various additives.

COF	0	2%	4%	6%	8%
EP + MoS_2_	0.63	0.32	0.26	0.22	0.23
EP + H-CNTs	0.63	0.32	0.32	0.38	0.49
EP + 6%MoS_2_ + H-CNTs	0.63	0.24	0.26	0.24	0.25

**Table 3 materials-12-00903-t003:** Nanoindentation data.

	HIT(O&P) (MPa)			E*(O&P) (MPa)		
	EP	EP + 6% MoS_2_ + H-CNTs	EP + 6% MoS_2_	EP	EP + 6% MoS_2_ + H-CNTs	EP + 6% MoS_2_
Data: 1	12.7	17.2	16.5	366.6	587.9	517.3
Data: 2	14.5	58.1	18.6	402.0	1034.1	518.9
Data: 3	26.8	23.3	17.5	756.5	642.4	487.2
Data: 4	12.7	28.6	18.0	566.0	681.3	508.5
Data: 5	20.5	19.4	17.7	589.1	618.2	529.1
Mean	17.4	29.3	17.6	536.0	712.8	512.2
Std Dev	6.1	16.6	0.8	157.3	182.8	15.8
Median	14.5	23.3	17.7	566.0	642.4	517.3

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
