# Peer review of "Tribological Properties of Molybdenum Disulfide and Helical Carbon Nanotube Modified Epoxy Resin"

_materials, 2019, doi:10.3390/ma12060903_

Reviewer 1 Report

The paper analyzes the antifriction and reinforcing action of two fillers (molybdenum disulfide and helical carbon nanotubes) against the epoxy matrix. The interesting aspect is that “ the composites with low friction coefficient, low wear rate, and high elastic modulus prepared in this study may be used as solid lubricants in high temperature and pressure environments where lubricants and greases cannot work.” ( line269-271) and abstract ( line 2-24).

The paper is well articulated and several investigations have been done to demonstrate the action of both fillers. However, there is no explanation of "why" the fillers give that particular effect, although through possible hypotheses. Moreover it lacks and/or convinces little how these fillers interact with the epoxy resin and why and if they interact with each other. We must remember that MoS2 and CNT are inorganic and non-polar materials, while the epoxy resin is organic in nature and has polar functional groups.

Furthermore, it is necessary to deepen the introductory part from the point of view of the effect of the filler inside the matrix and the consequent matrix properties improvement. The problem of the carbon nano tubes agglomeration needs various methods to increase their dispersion (similarly for Molybdenum disulfide). You see, for example:

1. Journal of Composite Materials (2014 )47(24):3091–3103 https://journals.sagepub.com/doi/full/10.1177/0021998312462431

2. Polymer Composites (2014) 37(4) https://onlinelibrary.wiley.com/doi/full/10.1002/pc.23260

3. Materials & design (2012) 39:432-436 https://www.sciencedirect.com/science/article/pii/S0261306912000969#bi0005

4. IOP Conf. Series: Journal of Physics: Conf. Series 991 (2018) 012054

https://iopscience.iop.org/article/10.1088/1742-6596/991/1/012054/pdf

This effect certainly influences both the physical and the final mechanical macroscopic properties of the material.

Important: Authors must check the References cited in the Introduction because there is no correspondence with the names of the cited Authors.

Reviewer 2 Report

The paper deals with the tribological and mechanical properties of epoxy resins (EP) modified by the addition of MoS2 and/or helical carbon nanotubes (H-CNTs). The study is interesting and well-structured. Some interesting findings regarding the use of MoS2 and H-CNTs as additives to EP are reported.

The paper is suitable for publication in the Materials journal; however, it currently has some issues which need to be addressed first. The more important ones are (1) argumentation of the selection of testing conditions for tribological tests, (2) detailed discussion of wear mechanisms by possibly providing images of the wear tracks and (3) presentation of results for the relevant samples (as described in the detailed corrections below).

Below a detailed list of suggested corrections including grammar corrections, writing improvements and other observations is provided. It is suggested that authors address these and perform the required corrections where necessary.

Detailed corrections

- Line 23: Instead of “solid lubricants”, the term “materials” would be more appropriate as “solid lubricants” is typically used to describe solid lubricating additives such as MoS2, PTFE etc.

- Line 30: For references [1,2], more general works regarding the properties of epoxy resin should be cited.

- Lines 43 to 47: References [16-18] are not cited correctly since the names of the authors do not correspond with the ones provided in the text.

- Line 50: In reference [20] coiled CNTs are considered and not helical CNTs as authors state.

- Line 68: Bubbles should be written with small initials and at the end of the sentence a point should be used instead of a semicolon: “…bubbles and water.”

- Line 70: Specify to which level (surface roughness) the samples were sanded.

- Lines 74-75: UMT friction tester is produced by the company Bruker, not Brooke.

- Section 2.2: The tribological tests were conducted over very short time periods (5 min) and sliding distances (1.5 m), which is unusual for tribological testing of sliding friction and wear. It is therefore questionable how the measured values correspond to the long-term friction and wear behaviour. It is suggested that the either: (a) authors make additional long-term tests (for at least 30 min) with the selected specimens to verify the correlation between the short-term and long-term tests or (b) to clearly emphasize the fact of short-term testing and its questionable transferability to longer time periods.

- Line 79: Instead of the term “scratch”, the use of the term “sliding” would be more appropriate for tribological tests. Scratch test is a different type of measurement.

- Line 82: Please rephrase the following part “…to ensure that its surface remained on the level” or describe in more detail what “on the level” means.

- Line 88: It should be “…to obtain the average friction coefficient.”

- Figures 1 and 2 are not very informative. It is suggested to use (additional) schematic images where the mechanisms of functioning also are presented.

- Section 3.1 could be moved to the Materials and Methods section since it described the properties of the samples.

- Line 94: Space is missing between “microscope)” and “image”.

- Lines 99-102: See comment to Fig. 4 (below).

- Fig. 3: For comparison, it would be useful if all subfigures (a-f) were also presented at the same magnification (e.g. in the corners of the current detailed images).

- Fig. 4: EDX maps should be presented for all samples. Currently, it is not clear to which sample the provided EDS maps correspond.

- Line 110: Omit “parallel” as it is not necessary.

- Line 112: Instead of “optimum addition of friction coefficient was selected”, it should be “optimum MoS2 concentration was selected”.

- Table 2: Please specify how the specified coefficients of friction were calculated (average value over the entire test?) or to which time in the tribological test they correspond. The same applies to Fig. 5d and Fig. 9.

- Fig. 5: Why was the same procedure as for EP + 6% MoS2 not also performed for EP + 4% H-CNTs? Or to put it differently, why no tests were performed with EP + 4% H-CNTs containing different amounts of MoS2?

- Line 126: Instead of “and then became stable.”, it should be “and the most stable.”

- Lines 127-129: The sentence “As MoS2 is a typical…” should be moved after the end of the sentence on lines 124-125.

- Line 128: Instead of “van der Waals force, to the layers”, it should be “van der Waals force, the layers”.

- Line 130: Instead of “coefficient stabilizes”, it should be “coefficient increases”.

- Lines 132-137: This part is unclear and/or illogical – please rewrite.

- Line 140: It should be “Fig. 5(d) also revealed that for more than 2% of additives the friction coefficient”

- Line 140: Please use “Fig.” instead of “Figure” and use the same denotation style throughout the text.

- Line 144: It should be “but with MoS2 it was the most obvious.”

- Line 148: Instead of “broken line diagram”, please use “wear volume curve”. This applies to the entire text and Figure captions.

- Line 161: It should be “initially decreased with the increasing H-CNT content and then increased at more than 4% of H-CNTs.”

- Line 162: It should be “structural integrity and stronger chemical bonds, which impart…”

- Line 164: Authors state that H-CNTs increase the fracture strength of the EP; however it seems that fracture strength did not play any significant role in his study. It is suggested to omit comments regarding the fracture strength or to put them into the context of the present work.

Figs 6-8: What were the main wear mechanisms during tribological tests (abrasion, ploughing, adhesion, etc.)? Did any material break-outs occur on the surfaces? Please comment on the wear mechanisms more in detail. Currently, they are described in a relatively superficial and speculative manner. It would also be useful if optical or SEM images of the wear tracks would be added to Section 3.3 in order to gain a better overview of the wear mechanisms.

- Line 166: Instead of “greater resistance to friction process”, “greater adhesion” would be more appropriate.

- Line 172: It should be “Fig. 8 shows wear volume curves and 3D wear profiles…”

- Line 174: It should be “wear rate initially decreased with H-CNTs content and then increased at more than 6% of H-CNTs.”

- Lines 175-176: Please omit “The wear reduction was attributable to the addition of H-CNTs” since this is evident already from previous descriptions.

- Line 177: Authors claim that at more than 6% of H-CNTs the additives “destroyed the cross-linking structure inside the EP”. Is the increased wear not correlated with the increased adhesion as described for Fig. 7? Please unify the descriptions of wear mechanisms for different samples so that they are related to each other. As already mentioned, images of wear tracks would help with the understanding of the wear mechanisms.

- Line 179: Authors claim that “the addition of MoS2 had little effect on wear”; however this is not completely true since for 2-6% of MoS2 wear also decreased. It would be better to say e.g. “the addition of MoS2 had a less pronounced effect on wear”.

- Line 188 (Fig. 9 caption): Instead of “with the same additive”, it should be “with different additives”.

- Line 191: Instead of “it was concluded”, it should be “it can be concluded”.

- Line 192: Instead of “without improving”, it should be “without significantly improving”.

- Line 193: Instead of “did not improve”, it should be “did not significantly improve”.

- Lines 193-194: Please omit “in order to improve these results” as it is not necessary here.

- Line 196: It should be “the composites with different amounts of H-CNTs”

- Lines 207-208: It should be “the composite material with 6% MoS2 and 4% H-CNTs provided minimum friction and wear.”

- Line 210: Omit “only” as it is not necessary here.

- Line 212: It should be “have not been discussed yet”.

- Line 213: It should be “This means”.

- Line 214: Instead of “idea” the term “hypothesis” would be more appropriate.

- Line 217: It should be “…and low wear loss, therefore they were selected.”

- Lines 218-220: Omit the sentence “Finally, whether H-CNTs…” since it is not necessary here.

- Line 224: Please provide a relevant reference for the “O&P theory” and initially write the “O&P” with the entire words.

- Table 3: Why there are no results presented for EP with only H-CNTs?

- Fig. 10b: On x-axis use “EP”, “EP+6%MoS2+H-CNTs” and “EP+6%MoS2” instead of “1”, “2” and “3”.

- Line 234: Instead of “it could” it should be “it can”.

- Line 238: Instead of “I was evident” it should be “it is evident”.

- Lines 239-240: It should be “In order to explore the mechanisms of improvement of elastic modulus of the EP composite with H-CNTs…”

- Lines 242-243: It should be “…it is clear that the H-CNTs tightly pulled together the fractured surface in the cracks of the composites, thus increasing their elastic modulus.”

- Fig. 11b: It should be specified which sample (amount of additives) is shown here.

- Line 251: Please omit “respectively” as it is not necessary here.

- Line 255: Please omit “then tends to become” as it is not necessary here.

- Line 255: It should be “but the wear rate decreases with 2% of MoS2 and then increases”.

- Lines 258-259: It should be “friction coefficient of the composites decreased first (for 2%), then stabilized, and subsequently increased rapidly (for more than 4%), while the wear rate decreased up to 4% of H-CNTs and then increased.”

- Line 266: Instead of “It was concluded” it should be “It was observed”.

- Line 267: Instead of “>500 MPa”, it would be more accurate to say “∼530 MPa”.

- Line 270: Instead of “solid lubricants”, the term “materials” would be more appropriate as “solid lubricants” is typically used to describe solid lubricating additives such as MoS2, PTFE etc.

Author Response

Round  2

Reviewer 2 Report

The authors addressed all the issues raised by the reviewer and have changed the manuscript accordingly. However, several grammatic and typing mistakes have remained which should be corrected before publishing the paper. It is suggested to also address these as described in the detailed correction propositions below.

Detailed corrections

- Line 17: It would be better “of the composites initially increased”.

- Lines 46-49: It would be better “reinforced EP with carbon fibers and the results show that compared to pure epoxy Young's modulus and tensile strength of epoxy/MWCNTs at 0.34 wt.% of CNTs are increased for 21.98% and 58.32%, respectively, and for epoxy/fiberglass/MWCNTs at 0.17 wt.% of CNTs raised for 1.05 and 1.17 times, respectively.”

- Line 53: It would be better “would have even better properties as that filled with CNTs.”

- Line 60: Title of the sub-section should be capitalised, i.e. “2.1. Materials”.

- Line 72: It should be “are so small (in the range of a few 100 nm)”.

- The first sentence on line 80 should be moved to the end of the paragraph – on line 83.

- Line 87: There is an unnecessary space between “2” and “2” in the numbering of the subsection.

- Lines 100, 184, 236: Please use “Fig.” instead of “Figure” throughout the entire text.

- Fig. 4 caption should be “Elemental distribution maps for the surface of EP+6%MoS2+6%H-CNTs.”

- Line 117: Instead of “scratches,” the term “wear traces” should be used.

- Line 119: Instead of “indenter”, the term “counter-body” should be used. This applies to the entire manuscript.

- Line 121: Instead of “degree of scratch”, the term “velocity” should be used and instead of “scratch time”, the term “test time” should be used.

- Line 121: Instead of “30 min”, it should be “300 s”.

- Lines 122-125: It should be: “The test time was initially set to 30 min, but it was observed that coefficient of friction will reach a plateau after around 50 s, and afterwards it will change very little, therefore the test time was set at 300 s.”

- Line 127: Instead of “spherical indenter”, the term “ball” should be used.

- Line 133: It should be “…to obtain the average friction coefficient.”

- Line 146: the word “table” should be capitalised, i.e. “Table”.

- Line 148: A point is missing at the end of the sentence.

- Line 171: It should be “as the content increases”.

- Line 175: It should be “is the same as described above”.

- Line 184: Instead of “Figure 6”, it should be “Figure 7”.

- Lines 192 and 205: The first word of the sentence should be capitalised, i.e. “Wear”.

- Line 199: A space between ability” and “[21]” is missing.

- Lines 202-203: Please omit “to friction process”.

- Lines 211-213: It should be “But with the increase in H-CNTs content, the cross-linking structure inside the EP matrix was destroyed, leading to an increase in wear”.

- In the caption of Fig. 9, use the same text as for Fig. 7 and 8 (only adapted to Fig. 9).

- Line 220: It should be “friction coefficients”

- Line 223: It should be “with different additives”

- For Fig. 11 please indicate the direction of sliding (either by using arrows and/or with a description in the figure caption).

- Lines 251-252: It should be “In order to study the wear mechanism, SEM micrographs of the wear tracks of the composite material with pure EP and the optimal content of each additive were taken, as shown in Fig 11.”

- Line 253: It should be “the SEM micrograph of the worn surface”

- Line 256: It should be “Fig. 11 (b) shows the SEM micrograph”.

- Lines 256-257: It should be “It can be seen that cracks perpendicular to the sliding direction are still present on the wear track, but there are significantly fewer wear debris.”

- Lines 259-260: It should be “Fig. 11 (c) shows the SEM micrograph”

- Line 260: Instead of “there were furrows in parallel to”, it should be “where furrows parallel to”.

- Line 261: It would be better “However, it was obvious”.

- Line 263: Instead of “avoid”, the term “prevent from” would be more appropriate.

- Line 265: Instead of “so”, the term “therefore” would be more appropriate.

- Lines 266: It should be “Fig. 11 (d) shows the SEM micrograph”

- Lines 268 and 273: Instead of “grinding surface”, the term “worn surface” should be used.

- Line 287: It would be useful to add “O&P theory (see description below).”

- Line 291: Instead of “as”, it should be “is”.

- Line 299: “Poisson’s ratio” is written with a different font and the word “Poisson’s” should be capitalised.

- Line 301: It should be “are the elastic modulus and Poisson’s ratio of the sample, respectively;”

- Line 302: It should be “are the elastic modulus and Poisson’s ratio of diamond head, respectively.”

- Line 341: It should be “of three typical specimens”.

Author Response

Dear Reviewer:

         Thank you for your comments. We have corrected all the mistakes you pointed out. The modification details are in the manuscript (in red color text).

Looking forward to hearing form you.

Thank you and best regards.

      YuYang

     2019/3/8